



# Evidence of eddy-related deep ocean current variability in the North-East Tropical Pacific Ocean induced by remote gap winds

Kaveh Purkiani[1], André Paul[1], Annemiek Vink[2], Maren Walter[1], Michael Schulz[1], and Matthias Haeckel[3]

[1]MARUM-Center for Marine Environmental Sciences and Faculty of Geosciences, University of Bremen, Bremen, Germany.
[2]Federal Institute for Geosciences and Natural Resources (BGR), Hannover, Germany
[3]GEOMAR Helmholtz Center for Ocean Research Kiel, Kiel, Germany

**Correspondence:** Kaveh Purkiani (kpurkiani@marum.de)

**Abstract.** There has been a steady increase of interest in mining of deep-sea minerals in the Clarion-Clipperton Zone (CCZ) in the eastern Pacific Ocean during the last decade. This region is known to be one of the most eddy-rich regions in the world. Typically, mesoscale eddies are generated by intense wind bursts channelled through gaps in the Sierra Madre mountains in Central America. Here, we use a combination of satellite and in situ observations to evaluate the relationship between deep-sea

current variability at the region of potential future mining and Eddy Kinetic Energy (EKE) in the vicinity of gap winds.

A geometry-based eddy detection algorithm has been applied to altimetry sea surface height data for a period of 24 years from 1993 to 2016 in order to analyse the main characteristic parameters and the spatiotemporal variability of mesoscale eddies in the North-East Tropical Pacific Ocean (NETP). Significant differences between the characteristics of eddies with different polarity (cyclonic vs. anti-cyclonic) were found.

For eddies with lifetimes longer than one day, cyclonic polarity is more numerous that anticyclonic rotation. However, anticyclonic eddies are larger in size, show stronger in vorticity, and survive longer in the ocean than cyclonic eddies (often 90 days or more). Besides the polarity of eddies, the location of eddy formation should be taken into consideration when investigating the impacted deep ocean region, as we found eddies originating from the Tehuantepec (TT) gap wind lasting longer in the ocean and travelling farther distances in a different direction compared to eddies produced by the Papagayo

(PP) gap wind. Long-lived anticyclonic eddies generated by the TT gap wind are observed to travel distances up to 4500 km offshore, i.e. as far as west of 110°W.

EKE anomalies observed in the surface of the central ocean at distances of ca. 2500 km from the coast correlate with the seasonal variability of EKE in the region of the TT gap winds with a time lag of 5-6 months. A significant seasonal variability of deep ocean current velocities at water depths of 4100 m was observed in multiple year time-series data likely

reflecting the energy transfer of the surface EKE generated by the gap winds to the deep ocean. Furthermore, the influence of mesoscale eddies on deep ocean currents is examined by analyzing the deep ocean current measurements when an anticyclonic eddy crosses the study region. Our findings suggest that despite the significant modulation of dominant current directions driven by the bottom-reaching eddy, the current magnitude intensification was not strong enough to trigger local sediment resuspension in this region. A better insight of annual variability of ocean surface mesoscale activity in the CCZ and their

effects on deep ocean current variability are of great help to mitigate the impact of the benthic ecosystem from future potential



deep-sea mining activities. On an interannual scale, a significant relationship between cyclonic eddy characteristics and El-Niño Southern Oscillation (ENSO) was found, whereas no robust correlation was detected for anticyclonic eddies.

## 1 Introduction

The CCZ holds the world's largest known contiguous resource of polymetallic (manganese) nodules on its deep ocean floor (Beiersdorf, 2003). Economic interest for mining of these nodules has led to a rapid increase in the number of exploration licenses that have been granted for by the International Seabed Authority (ISA), totalling 16 contracts in the CCZ licenses in 2019 and covering about 1.2 million square kilimeters of seafloor. Potential future deep-sea mining (DSM) activities will inevitably produce a plume of suspended sediment above the seafloor. The footprint of the suspended sediment plume and its ecological impact on the pelagic and benthic fauna will depend on several factors such as the bottomwater current regime, seafloor topography, sediment particle size distribution and concentration, and particle settling rates (e.g., Gillard et al., 2019). Current variability at the seafloor has been shown to be closely related to the passage of mesoscale eddies in the CCZ (Aleynik et al., 2017). These authors postulate that mining-related sediment plumes could spread more widely and rapidly during eddy-induced elevated bottomwater current flow periods. Thus, understanding the long-term characteristics of eddies in the CCZ and their effects on abyssal current variability and plume behaviour can be pivotal for developing mitigation measures to minimise the spatial scale of mining impacts on the seafloor.

Advances in high spatio-temporal resolution of sea surface height measurements from satellite altimeters have enhanced our knowledge of ubiquitous features in the world's oceans, such as mesoscale eddies. Mesoscale eddies are large bodies of swirling water typically with radii in the order of 100 km and with lifetimes ranging from days to a few months (Chelton et al., 2007). Their sizes vary with latitude, bottom topography and the nature of their generation (Rhines, 2004).

The NETP is known as one of the most eddy-rich regions, typically at the mesoscale, in the world ocean (Fiedler, 2002; Chelton et al., 2007). Understanding eddy genesis processes in the NETP is very complex and is known to be different from other basins in the tropical Pacific (Hansen and Paul, 1984). The intense wind burst, channelled through gaps in the Sierra Madre mountains in Central America, is known to be the main reason for eddy genesis and propagation in this region (Chelton et al., 2000). Strong winds occur when the northern mid-latitude cold fronts penetrate into the American tropics in winter in association with high atmospheric pressure over the Gulf of Mexico and a strong pressure gradient across the isthmus (Hansen and Paul, 1984; Romero-Centeno et al., 2003). However, not all of the observed eddies in the NETP can be explained by the effect of gap winds. Other generation mechanisms including Ekman pumping and conservation of potential vorticity as the North Equatorial Counter Current being deflected north by the coast to form the Costa Rica Coastal Current are known to generate mesoscale eddies in this region (for more details see Hansen and Maul, 1991; Willett et al., 2006). Moreover, the numerical study of Liang et al. (2012) shows that neglecting Kelvin wave forcing in generation of mesoscale eddies in NETP causes under-estimation of mesoscale variability by 6-12% in the Tehuantepec region and by 2-13% in the Papagayo region.

Mesoscale eddies in the ocean interior are more energetic than the surrounding water and are the influential component of dynamic oceanography in the oceans. They can interact with background ocean currents and thus play a key role in the





anomalous transport of momentum (Farneti et al., 2010; Hill et al., 2015), sediment (Washburn et al., 1993; Zhang et al., 2014;
Aleynik et al., 2017), heat (Lyman and Johnson, 2015), oxygen (Stramma et al., 2014; Czeschel et al., 2018) and nutrients
(Müller-Karger and Fuentes-Yaco, 2000; Liang et al., 2009) into the ocean interior.

Most of the previous studies of mesoscale eddies in the Pacific Ocean are focused on the surface processes and little is known
about the impact of mesoscale eddies in the deep ocean, specifically on their current characteristics (e.g., for the northwestern
Pacific see Lee et al. (2013); Xu et al. (2019), for the southwestern Pacific see Qu and Lindstrom (2001); Keppler et al. (2018);
Liu et al. (2012) and for the southeastern Pacific see Chaigneau and Pizarro (2005); Chaigneau et al. (2008); Stramma et al.
(2014); Thomsen et al. (2015)). The study of Stramma et al. (2014), however, had a particular focus on investigating the role
of mesoscale eddies on the ocean current characteristics. Their study shows that an extremely long-lived anticyclonic eddy
carried an anomalous, oxygenated water mass from the Chilean coast to the open ocean with a mean propagation velocity of
$5.5 \ \mathrm{cm s^{-1}}$ and stayed isolated during the 11 months of its travel time. The authors also identified the eddy signature in current
velocity observation recorded by current meter instruments between 13 and 601 m depth.

Recently, eddy detection and tracking algorithms using altimetry data seem to be an extremely useful tool to study mesoscale
eddies (Chelton et al., 2007; Chaigneau et al., 2008; Nencioli et al., 2010; Dong et al., 2012). Employing such algorithms was
successful in characterizing different aspects of mesoscale eddies such as size, intensity, track, translation velocity, and lifetime.
Despite of all advances in satellite altimetry observation and eddy detecting algorithms, most of our knowledge about mesoscale
eddy characteristics in the NETP is limited to only few studies which have been carried out in the last decade (Müller-Karger
and Fuentes-Yaco, 2000; Willett et al., 2006). It has been reported that between 4 and 18 eddies are formed annualy in the
NETP during the boreal winter. It has been reported that anticyclones are more numerous, larger and last longer than cyclonic
eddies in this region.

Many aspects remain largely unknown especially the eddy characteristics in the ocean interior and their impact on the deep
ocean current properties. Furthrmore, we still do not know, for example, how far do the eddies travel into the ocean? How
do surface mesoscale eddies impact the deep ocean current properties? Is the variability of eddy characteristics related to
large-scale climate variability such as El-Niño Southern Oscillation (ENSO) in this region? The understanding of the role of
TT and PP gap winds in the generation of long-lived mesoscale eddies in this region is another interesting issue as the gap
winds have different periods of annual activity. Due to potential future DSM in the NETP, the response of deep ocean current
characteristics to the passage of surface mesoscales is of interest in order to assess its impact on the spatial and temporal
footprint of the sediment plume generated by mining and the possibility to resuspend the freshly deposited sediment blanket.

The main goal of this study is to identify mesoscale eddies using the SSHA data and to compare the main characteristic
parameters of cyclonic and anticyclonic eddies in the NETP. Secondly, we focus on the physical response of deep ocean
current properties to passage of surface mesoscale eddies to elucidate the vital role of mesoscale eddies on sediment dispersal
in connection of future potential DSM activity in the CCZ.





## 2 Data and methods

### a. Data

The combinations of satellite altimetry data, deep ocean current velocity measurements and a set of reanalysis model products were used to identify eddies and to explore the impact of long-lived eddies on the deep-sea environment in the

NETP. AVISO data (Archiving, Validation, and Interpretation of Satellite Oceanographic data; see www.aviso.oceanobs.com) have been successfully applied in previous studies to identify mesoscale eddies and track them in different basins (e.g., Palacios and Bograd, 2005; Chaigneau et al., 2008; Dong et al., 2012). Thus, the altimetry products of merged daily mean sea surface height data was obtained from AVISO for this study, with 1/4° grid spacing from 1993 to 2016, in order to understand the spatiotemporal evolution of mesoscale eddies. To obtain the sea surface height anomaly (SSHA),

the climatological sea level in this period was subtracted from the corresponding sea surface height.

A set of long-term current measurements is used which is obtained from just above the seafloor by Ocean Bottom Mooring (OBM) systems that were deployed in the northeast Pacific Ocean in a potential future DSM site between July 2013 and April 2016. Three moorings were deployed at the geographical position of 11.5°N, 117.1°W and at a water depth of ca. 4100 m in a triangle with a distance of 8 km between the moorings. Each mooring was equipped

with an upward looking Acoustic Doppler Current Profiler (ADCP) that measured the velocity and direction of ocean currents in the lower 40 m of the water column with bin sizes around 2 m starting from 7 m above the seafloor with a sampling interval of 60 min during the first year and 45 min during the following 3 years. Moreover, a short term array of single-point current meters attached to a thermistor chain including three Aquadopps at different depths of 6 m, 206 m, and 406 m above seafloor was deployed from 20 March to 02-June 2015 in a location with a depth of 4180 m at the

potential DSM region (see Figure 10e). The ocean current measuring array is located at the center of the OBM triangle and recorded data at every 150 s which provides us the best available estimates of current properties of deep ocean at the last 400 m above the seafloor in this region.

Finally, an eddy-resolving global ocean reanalysis products with a horizontal resolution of 1/12° and 50 vertical layers covering the period between 1993 and 2016 was used for further comparison and examination of hypothesises in our

study (Drévillion et al., 2018).

### b. Eddy detection algorithm

The automated eddy detection method developed by Nencioli et al. (2010) was applied to the data to quantify mesoscale eddies in the NETP. The eddy-detecting algorithm operates on the basis of the following four principal attributes:

1) the zonal velocity component has to change its sign along a meridional section across the eddy centre and its

magnitude has to increase away from the centre;

2) the same as condition (1) but the meridional velocity has to change its sign along a lateral section across the eddy centre;





3) the eddy centre has a minimum local velocity;

4) the direction of the velocity vectors should follow the same direction of rotation.

The rotating bodies of water are detected and are taken as eddies when all conditions above are satisfied. The potential center of an eddy is identified when the conditions are satisfied by all the vectors along the baoundaries of the searching area, there is a closed circulation around the velocity minimum; therefore the point is recorded as a center of an eddy. Closed streamlines are computed surrounding each detected eddy center, with the area enclosed in the outer streamlines defined as the eddy region. If an eddy has been successfully detected at time $t$, the same type of eddy (cy-

clonic/anticyclonic) at time $t+1$ is checked in the search region, defined by two dimensionless arbitrary parameters $a$ and $b$. The size of the search area strongly affects the accuracy of the detecting algorithm in this method. An optimal combination of $a=4$ and $b=3$ was set in this study due to higher success of detection rate and lower excess of detection rate (Nencioli et al., 2010). Another important limit which may cause inaccuracy in detecting eddies is the splitting of a long-life eddy into two or more distinct eddies. To reduce the chance of this type of error, if a centre cannot be located

within the search area at $t+1$, a second search will be employed at $t+2$ with half of the radius in $t+1$. This will avoid merging eddies with different centres, specifically when the search radius is too large or the grid points are too coarse. For further detailed information on the eddy detection algorithm, readers are referred to Nencioli et al. (2010).

## 3 Results

### 3.1 Seasonal variability of Eddy Kinetic Energy (EKE)

The EKE as a measure of mesoscale eddy activity is calculated from Sea Surface Height Anomalies (SSHA) derived from high-pass filtered satellite products as follows:

$$EKE = \frac{1}{2}\big(u_g'^{\,2} + v_g'^{\,2}\big),\tag{1}$$

where $u_g'$ and $v_g'$ are deviations from monthly mean geostrophic velocities, which are computed from SSHA gradients as

$$u_g' = -\frac{g}{f}\frac{\partial(\text{SSHA})}{\partial y}, \; v_g' = \frac{g}{f}\frac{\partial(\text{SSHA})}{\partial x},\tag{2}$$

where $g$ is the gravitational acceleration, $f$ is the Coriolis parameter, and $x$ and $y$ are the eastward and northward distances (Stammer, 1997). The spatial distribution of mean seasonal EKE constructed from the 24-yr SSHA data in the NETP for the time period between 1993 and 2016 is shown in Figure 1.

In addition to the region of the North Equatorial Counter Current (south of 6°N) where the EKE permanently shows values higher than $500\,\mathrm{cm^2\,s^{-2}}$, a greatly elevated EKE signal is located in the region of the gap winds off the coast of Mexico, albeit

with a strong seasonal variability. An EKE with values larger than $700\,\mathrm{cm^2\,s^{-2}}$ is found in fall and winter, when strong gap wind events initiate ageostrophic ocean currents (see Figure 1a-d). In the spring season (Figure 1b), the EKE reduces to values of about $400\,\mathrm{cm^2s^{-2}}$ in the Tehuantepec (TT) and Papagayo (PP) gulfs. In summer, the intensity and occurrence of northerly





winds in the TT gap wind region is reduced (Romero-Centeno et al., 2003), thus the EKE drops to values below 300 $\mathrm{cm^2s^{-2}}$ and its effect on the background EKE of the ocean is restricted to a much smaller region (see the white contours in Figure 1c).

A comparison of seasonal variation of EKE in the NETP reveals an interesting feature. Despite the lower amount of EKE in April-June with no significant peak, the energetic region is pushed further offshore in this period (see the position of the white solid line in Figure 1). A bi-seasonal variation of EKE in the region of the PP and TT gap winds in summer and autumn is also evident. The summer peak in EKE in the PP gap wind region is due to an increase in easterly winds during August in this region (Romero-Centeno et al., 2003).

## 3.2 Eddy detection, numbers and lifetimes

An example of the application of the eddy detection algorithm on SSH data is shown in Figure 2. On the 31-01-2016, 15 eddies with different size, strength and shape are shown as closed black contours. Sea level height and associated geostrophic currents are shown in the background throughout the entire studied region (NETP). The centre of each eddy, which is defined as the location with minimum local velocity, is depicted as a black star. The current fields are shown as by the arrows and the direction

of current rotation reveals the polarity of eddies.

Applying the eddy detection geometry algorithm on the long-term SSHA data from 1993 to 2016 in the NETP enables us to extract some important information on eddy characteristic parameters such as abundance, size, lifetime, polarity, translation and swirl velocity. Any eddy that satisfies the eddy detection algorithm attributes and lasts for longer than 2 days is counted as a significant eddy in our analysis. An eddy is not counted at each time step of its lifetime, but is dealt with as a single eddy (for

its entire lifetime) from the time of formation until its decay. In total, 6202 cyclonic (CE) and 5363 anticyclonic (ACE) eddies were detected. This can be translated to a mean annual number of 258 CEs and 223 ACEs in this region. Thus, the total number of CEs exceeds the ACEs by about 16%. It should be noted, however, that the number of detected eddies very much depends on the arbitrary parameters defined earlier in section 2. Most of the detected eddies have a lifetime shorter than 7 days. Figure 3 shows the distribution of eddy abundance when eddies with lifetimes shorter than 7 days are excluded. ACEs with lifetimes

longer than 50 days occur frequently, whereas CEs with such lifetimes are uncommon. Eddies of both types have peak lifetimes of around 14 days. The average lifetimes of CEs and ACEs over the period of our study are 18 days and 28 days, respectively, meaning that ACEs last almost 50% longer in the ocean than CEs. The average lifetime of ACEs detected in the NETP is in the same order as those described from the southeastern Pacific Ocean (Chaigneau et al., 2008), although the lifetime of CEs in the NETP is shorter than CEs in that region. Eddy analysis in the subtropical zonal band of the northern Pacific Ocean (Liu

et al., 2012) has revealed much longer eddy lifetimes, resulting from the instability of the Kuroshio current. Though in the same basin of the Pacific Ocean, Dong et al. (2012) show that eddies in the southern California Bight have relatively short lifetimes, with an average lifetime of 5-14 days, comparing well to our results. Varying origins, advection processes and decaying factors might be the reason for the different eddy lifetimes observed in the various basins of the Pacific Ocean.

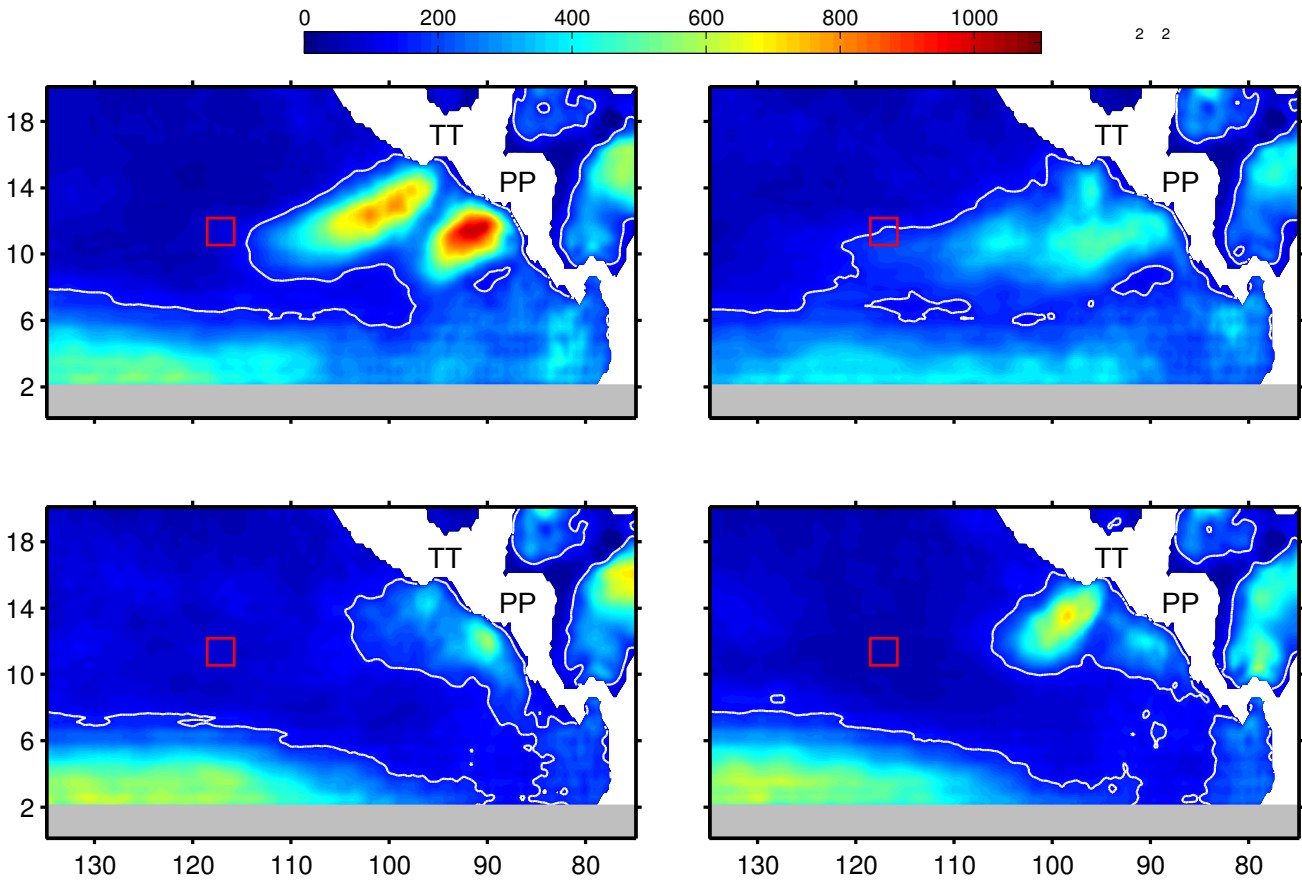

**Figure 1.** Spatial distribution of seasonal mean Eddy Kinetic Energy (EKE) over a period from 1993 to 2016 obtained from AVISO satellite altimetry data, for a) Jan-February-March, b) April-May-June, c) July-August-September, and d) October-November-December d). The solid white contours represent an EKE of 150 $cm^2s^{-2}$, which is assumed as a threshold for highly energetic regions. The red square indicates the location of the study region (abbreviated as SR in the text).



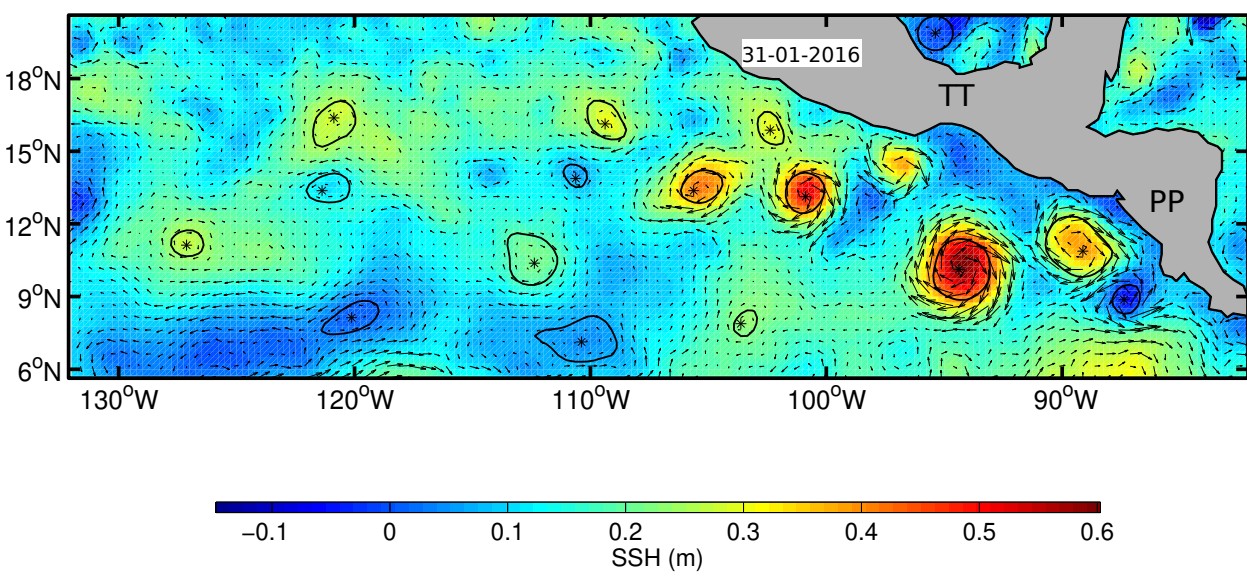

**Figure 2.** An example for applying the eddy detection algorithm on sea surface height altimetry data for the 31-01-2016. Detected eddies are shown as closed black contours. The associated geostrophic field is shown by the black arrows. Cyclonic and anticyclonic eddies can be identified as differences in direction of water mass rotations. Centres of eddies are indicated as the black stars for each individual vortex. The amplitude of sea surface height is colour-coded.

**Figure 3.** Number of a) anticyclonic eddies (ACE) and b) cyclonic eddies (CE) from 1993-2016 plotted as a function of their lifetime. Eddies with lifetimes shorter than 7 days were excluded from the analysis. The binwidth for eddy lifetime is 7 days.





### 3.3 Spatial distribution of eddies

Identifying regions of eddy genesis and high eddy abundance is important for understanding eddy lifetimes and trajectories. The zonal variability of meridionally averaged EKE over the entire period of 24 years between 9°N and 16°N is shown in Figure 4. The figure shows that regions with high EKE are restricted to the continental shelf, with a rapid decrease towards EKE values in the open ocean. Two significant peaks are observed in the EKE values, which correspond geographically to the zonal location of the Tehuantepec and Papagayo gap winds. These are thus hotspots for eddy genesis. No eddy formation

hotspot is found offshore. Moreover, west of 110°W the EKE and its variation is reduced by a factor of 5, showing that the energy level in the open ocean cannot be satisfactorily explained by offshore eddy formation. As eddies propagate into the open ocean, they encounter the East Pacific Rise (EPR) at ca. 105°W. The EPR is a long, north-to-south oriented mid-ocean ridge system whose height is about 1200 m shallower than the surrounding seafloor. The rapid drop in EKE west of 105°W is most likely caused by this topographic feature (Palacios and Bograd, 2005).

In order to further explore the spatial distribution of eddies in the NETP, the probability of eddy occurrence (CE vs. ACE) was calculated using the available data from 1993 to 2016, with a horizontal resolution of 1° (Figure 5a-b). The probability of eddy occurrence is defined at each cell as the percentage of time that the cell is located within a vortex. The eddy probability ranges between 0 and 68% for anticyclonic and 0 and 62% for cyclonic eddies. The coastal regions close to the gap winds show the highest probability for both types of eddies with values larger than 30% (compare the number of cells with high probability

in Figure 5a-b). Moreover, the region with high eddy probability for ACEs is extended far offshore west of 100°W, while for CEs the region of high eddy probability is limited to the east of 100°W.

### 3.4 Eddy Vorticity and Radius

We define the vertical relative vorticity ($\zeta = \frac{\partial v}{\partial x} + \frac{\partial u}{\partial y}$) of an eddy at its center. A histogram of the relative vorticities for cyclonic and anticyclonic eddies with lifetimes longer than 7 days is shown in Figure 6a. Generally, ACEs have higher absolute values of

relative vorticity than CEs. The peak of absolute vorticity for both types of eddies is between $1.5 \times 10^{-6}$ s$^{-1}$ and $2 \times 10^{-6}$s$^{-1}$, but with stronger asymmetries towards eddies with higher vorticities for ACEs. The normalised distribution of ACEs is more skewed to the larger vorticities compare to the CEs. The mean vorticity of all ACEs at entire period of time (-0.26×10$^{-5}$) also has a larger magnitude than that for all CEs (0.2×10$^{-5}$).

As anticyclonic eddies move westward from the coast of Mexico, they encounter the East Pacific Rise with a change in

seafloor depth from 4000 m to 2800 m. In order to conserve the potential vorticity as the depth decreases, the absolute vorticity ($\zeta + f$) must also decrease. Therefore, eddies with strong intensity deviate toward the equator and lose their planetary vorticity. Our analysis shows that the relative vorticity of ACEs for each degree of meridional displacement reduces by a factor greater than $3.5 \times 10^{-6}$(s$^{-1}$). For the weaker eddies that do not have enough energy to move equatorward, the topographic blocking might be the main reason for decay of eddies due to the combined effects of reduced relative vorticity and increased frictional

dissipation of eddy kinetic energy caused by the elevated seafloor.

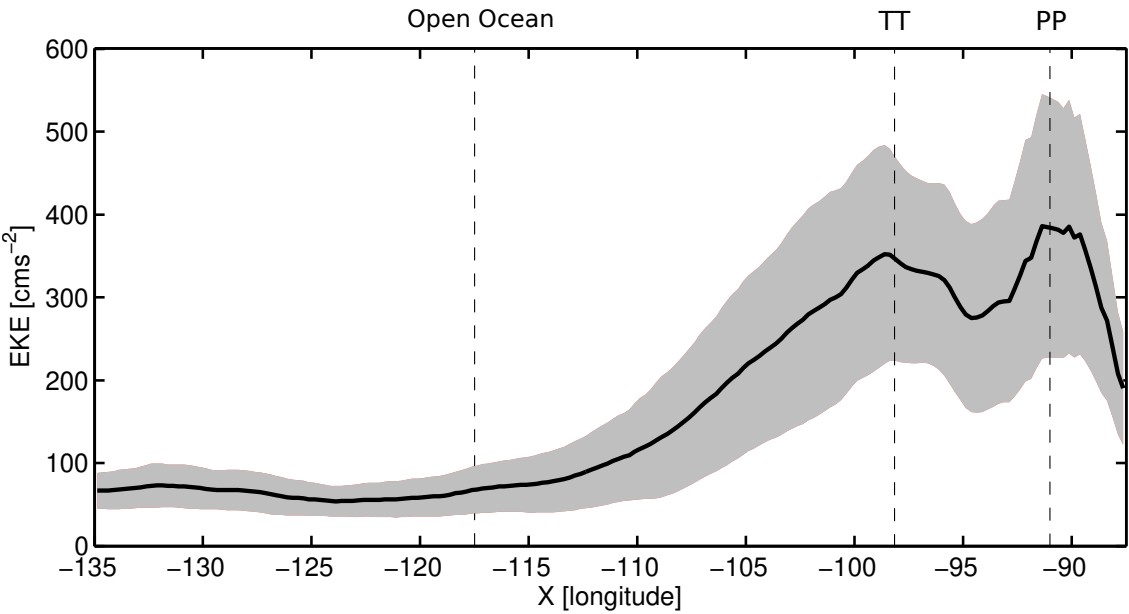

**Figure 4.** Zonal variability of meridionally averaged EKE between 9°N and 16°N). The values are then averaged over the period period of study from 1993 to 2016. The shaded area shows the ± 0.5 of standard deviation averaged over the whole period at each position. The dashed lines show the location of opean ocean, TT and PP respectively. The continental shelf is on the right side of the figure.

**a) ACE**

**b) CE**

**Figure 5.** Spatial distribution of the probability of eddy occurrence in bins of 1° by 1° resolution for a) anticyclonic eddies (ACE) and b) cyclonic eddies (CE).







**Figure 6.** (a) Histogram of eddy vorticity for eddies with a lifetime longer than 7 days for the time interval between 1993 and 2016 with a binwidth of 0.05. Cyclonic (CE) and anticyclonic eddies (ACE) are mirrored for better visualisation. The eddy vorticity is normalised to the local Coriolis parameter. (b) Histogram of eddy radius for anticyclonic eddies with lifetimes longer than 7 days during the time interval from 1993 to 2016. (c) Histogram of eddy radius for cyclonic eddies with lifetimes longer than 7 days during the time interval from 1993 to 2016. The number of eddies have been normalized to maximum number of anticyclonic eddies (736 ACEs with radius of 80 km). The thick grey line depicts a normalized number of 0.5.

As eddies are not necessarily circular in shape, an eddy radius is defined as $R = \sqrt{\frac{A}{\pi}}$, where $A$ is the area delimited by the eddy's edges.





Understanding the size of eddies in the ocean can help us to analyze the eddy intensity and determine their impact on the open ocean. Histograms of eddy radius distribution for CEs and ACEs with lifetimes longer than 7 days during the time
interval between 1993 and 2016 are shown in Figure 6b-c. The number of eddies of each type are normalized to the maximum abundance accordingly.

A non-Gaussian distribution of eddy radius with strong asymmetries is observed for both types of eddies. The radius of ACEs with a normalised number higher than 0.5 (shown by the dashed black line in Figure 6b) shows a wide range from 60 to 110 km. In contrast, CEs are limited to a radius between 60 and 90 km.

A high number of ACEs have radii between 60 and 110 km. The abundance of eddies of both types with a large radius (>150 km) are infrequent but cannot be neglected. The temporal mean eddy radius between 1993 to 2016 for ACEs is 92 km, whereas CEs have a smaller mean radius of 84.5 km. Further analysis on the radius of ACEs obtained from year to year between 1993 and 2016 show only a small long-term variation in eddy size. The small changes in long-term standard deviation of eddy radius (32 km± 2.5 km) illustrate that decadal climate variability has a negligible impact on eddy size in this region. However, the
large value of mean annual standard deviation of eddy radius (32 km) indicates inhomogeneity in size of eddies, reflecting a strong seasonality in eddy formation in this region.

Besides the seasonal variability in the size of eddies, the analysis of spatial distribution of eddy radius in this region shows that the size of eddies increases with decreasing latitude. This is associated with the baroclinic Rossby deformation radius changing with latitude from 60 km at 20°N to 150 km at 6°N (Chelton et al., 1998).

## 3.5  Translation speed and swirl velocity of eddies

The average translation velocity of an eddy is determined by considering the distance between the location of eddy generation and eddy degeneration and the time travelled by the eddy. Significant differences are found between the averaged translation velocities of ACEs and CEs. The analysis of all ACEs with lifetimes longer than 7 days indicates that the average translation speed of ACEs is 12.5 $\mathrm{cm\,s^{-1}}$, with a minimum and maximum speed of 3.4 and 18.1 $\mathrm{cm\,s^{-1}}$, respectively. The high translation
velocity of ACEs is attributed to the large southward velocity component of eddies. In contrast, CEs show notably slower translation speeds, varying between 4.1 $\mathrm{cm\,s^{-1}}$ to 10.72 $\mathrm{cm\,s^{-1}}$ with a mean translation speed of 6.8 $\mathrm{cm\,s^{-1}}$.

Previous analyses of eddy translation speeds in the zonal band of the North Pacific Ocean (Liu et al., 2012) and in the Chile-Peru basin (Chaigneau et al., 2008; Stramma et al., 2014) indicate a lower mean translation speed of 4-7 $\mathrm{cm\,s^{-1}}$ for ACEs and CEs with almost no difference between the two eddy types. Therefore, the higher mean eddy translation velocity found for
ACEs in this region can be considered as a specific characteristic of eddies in the NETP. The mean translation speed of CEs in the NETP is very similar to those reported in previous studies mentioned above.

## 3.6  Trajectory of long-lived eddies

We consider eddies with lifetimes equal to or longer than 90 days to be long-lived eddies. From 1993 to 2016, 106 long-lived ACEs and only 7 CEs were detected in this region. The number of long-lived ACEs substantially exceeds the CEs detected in





**Table 1.** Mean distance travelled, mean translation velocity, and number of long-lived eddies per age class as shown in Figure 7

| Eddy Category | abundance | lifetime (days) | $\overline{distance}(km)$ | $\overline{translation\,velocity}(cm/s)$ |
|:---:|:---:|:---:|:---:|:---:|
| # 1 | 43 | 91-129 | 1500 | 16.17 |
| # 2 | 24 | 130-166 | 2120 | 16.58 |
| # 3 | 21 | 167-206 | 2680 | 17.09 |
| # 4 | 11 | 207-244 | 3340 | 17.24 |
| # 5 | 6 | 245-282 | 4150 | 18.50 |
| # 6 | 1 | 283-321 | 4570 | 16.49 |

the NETP which is different from the distribution of eddy abundance when all eddies longer than a day were considered (see section 3.2).

The amount of long-lived CEs presents a challenge to our statistical analysis and therefore, they were removed from our trajectory analyses. The trajectories of ACEs with a lifetime longer than 90 days are shown in Figure 7. To ease the comparison, eddies are divided into different lifetime classes between 91 and 321 days, with a time interval of 37 days.

The number of ACEs with lifetimes between 91 and 129 days almost equally derive from both the Tehuantepec (TT) and Papagayo (PP) gap wind regions, with a slightly larger number of eddies formed in PP than in TT. The number of eddies formed in the TT and PP gap wind regions for the next two lifetime classes, 130-166 days and 167-206 days, do not show significant differences. For the eddies with lifetimes longer than 207 days (eddies depicted with yellow, orange and red colors in Figure 7), the eddy generation region is different, with all eddies originating in the TT gap wind region. The genesis of long-lived

eddies in the TT gap wind region is consistent with the more energetic Tehuantepec jet winds (both in magnitude and duration) compared to the Papagayo jet winds (McCreary et al., 1989; Chelton et al., 2000; Romero-Centeno et al., 2003).

Long-lived eddies have a preferred range of eddy size. Both small and very large eddies do not last long in the ocean and dissipate due to their low intensity (Figure 8a). Most of the long-lived eddies have a vorticity range between $0.2\times10^{-5}\,\mathrm{s}^{-1}$ and $0.4\times10^{-5}\,\mathrm{s}^{-1}$ and a radius between 60 and 150 km (Figure 8a and b). An inverse relationship between eddy size and eddy

vorticity particularly for long-lived eddies is found. The eddy size appears to decrease when the vorticity of long-lived eddies increases.

A linear increase of eddy propagation distance with lifetime is seen for long-lived eddies (Figure 8d). In most cases, long-lived eddies travel for a distance of more than a thousand kilometres into the interior ocean (Table 1). In fact, long-lived eddies generated in the TT gap wind region can travel a distance of about 4500 km, which is almost twice as far as the travel distance

observed for eddies that originate in the PP gap wind region (Figure 7). Furthermore, except for the eddy category #6 which is the longest-lived eddy observed in the studied region, the mean translation velocity of eddies increases with lifetime (Table 1).





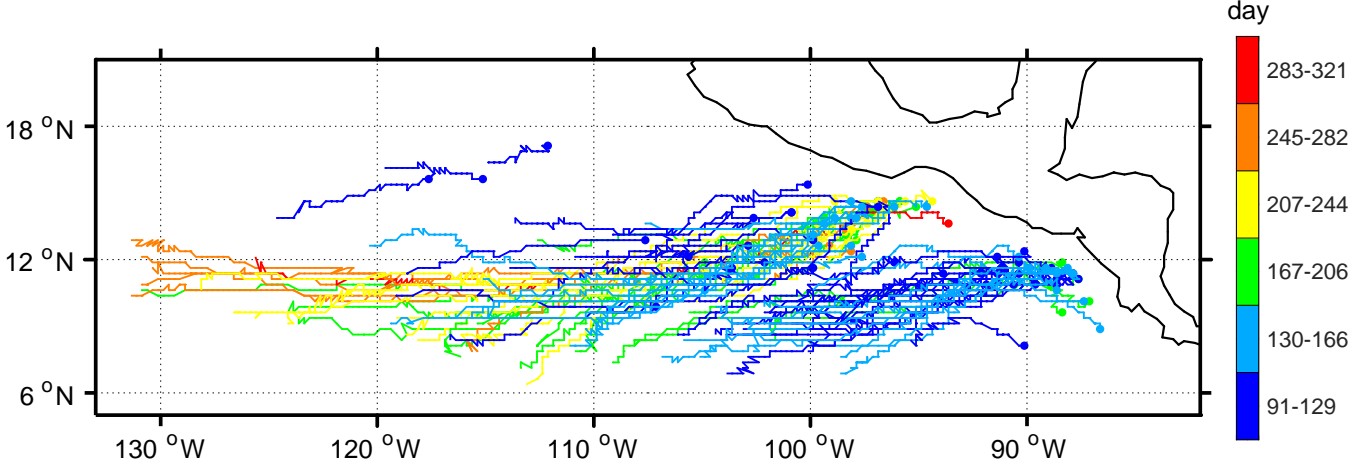

**Figure 7.** Trajectories of long-lived anticyclonic eddies (ACEs) with a lifetime equal to or longer than 90 days during the time interval from 1993 to 2016. Lifetimes have been divided into six classes as shown in the colour bar. Each colour indicates a lifetime with a length of 37 days.





**Figure 8.** Relationships between eddy lifetime and eddy radius, eddy vorticity, EKE, and propagation distance into the ocean for all anticy-clonic eddies with lifetimes longer than or equal to 7 days. Long-lived eddies are shown using the colour scheme as used in Figure 7. Eddies with lifetimes shorter than 90 days are depicted as grey circles. Note that each circle represents the mean value of every parameter throughout its life period against the eddy lifetime.



### 3.7 Interannual variability of eddy parameters

In the eastern Pacific ocean, the El Niño-Southern Oscillation (ENSO) is the most dominant climate mode on interannual timescales, and this could well have an effect on the variability of eddy characteristics in the NETP. To remove effects of
seasonal fluctuations on eddy properties, the annual mean of each characteristic eddy parameter was compared with the Oceanic Niño 3.4 Index (ONI) for the time period under consideration(1993-2016). The study period has clear phases of warm (cold) events associated with El Niño (La Niña) cycles.

Figure 9a shows the relationship between the ocean EKE in the TT and PP gap wind regions and the ONI. The EKE of the ocean at PP appears to be strongly related to ONI, and thus to large-scale climate variability. These findings are confirmed
by sea surface temperature anomalies in the gap wind regions (Alexander et al., 2012). The amount of wind energy induced to the ocean at PP increases during warm episodes of the ENSO cycles due to equatorward movement of the Inter-Tropical Convergence Zone (ITCZ), which enhances the positive wind curl on the southern flank of the PP wind jet (Karnauskas et al., 2008). In contrast, the wind energy induced to the ocean at TT is primarily modulated by another large-scale intra-basin mechanism called the Atlantic Tripole Pattern (ATP). This can be further demonstrated by comparing chlorophyll-a (CHL)
concentration at TT and PP gap winds during the famous 1997-1998 El Niño during which the wind stree was weaker (see Figure 5 in McClain et al., 2002). The lower level of CHL concentration at PP shows a strong relation with weaker wind stress. While, no correlation between these two parameters at TT gulf, confirms weaker influence of El Niño on TT gap winds.

The number of eddies, both for ACEs and CEs, shows a clear but weak positive relationship to the ONI (Figure 9b). Strong La Niña events (ONI <-0.5) apparently induce a decline in the number of eddies from both types. An increase in the number
of eddies was evident for the case of strong El Niño events (ONI > 0.5). The effect of weaker ENSO cycles with -0.5 < ONI < 0.5 on the variability of eddy number is more complicated and does not show a clear trend.

The relationship between size and interannual climate variability is different to that of eddy number. The correlation analysis shows a negative dependency of -0.41 between the radius of CEs and ONI. This means that during La Niña (El Niño) events, larger (smaller) CEs are generated in the ocean. The very weak correlation between the radius of ACEs and ONI (0.21) implies
no meaningful relationship in this case.

The eddy vorticity of CEs has been found to follow the ENSO variability with a positive and significant correlation of around 0.7. During a strong La Niña period, the vorticity of CEs decreases to the lowest value(Figure 9d). Eddy vorticity reaches its peak value during strong El Niño phases. This may explain the inverse relationship of ONI and size of CEs, assuming the conservation of angular momentum for a vortex in the ocean.

The correlation of eddy lifetimes and ONI show a negligible (-0.15) and a weak (-0.25) relationship respectively for CEs and ACEs, which in both cases are inverse (Figure 9e). This may indicate that ACEs live longer during La Niña phases, while a strong EL Niño event does not have significant impact on the lifetime of ACEs in the ocean.

The eddy intensity (EI), which is calculated as the mean EKE over the vortex area, shows a different behaviour for eddies with distinct polarity. Although the CEs show a moderate positive correlation (0.44) with ENSO cycles, the intensity of ACEs





demonstrates a negative low relationship with ONI. This means that the highest intensity of ACEs (CEs) takes place when strong La Niña (El Niño) events develop (Figure 9f).

In summary, the strength and direction of relationship between ONI and characteristic parameters for CEs illustrate that ENSO cycles have a profound effect on their development in the NETP. In contrast, ACEs are less related to ENSO cycles because they are mostly generated in the region close to the TT gap wind region and are not regulated by ENSO cycles
(Karnauskas et al., 2008).

## 4 Discussion

### 4.1 Lag response of ocean bottom current properties to an anticyclonic surface mesoscale eddy

Large eddies passing through the open ocean can have a profound effect on the variability of near-bottom currents even at depths of 4000 m or more (e.g. Demidova et al. (1993); Kontor and Sokov (1994); Liang and Thurnherr (2012); Zhang et al.
(2014); Aleynik et al. (2017)). To illustrate the response of the deep ocean environment to ocean surface mesoscale eddies, we look at the changes of current properties from 19-March to 02-June 2015 in the vicinity of the seafloor while an anticyclone eddy passes through the SR (Figure 10).

An anticyclonic long-lived eddy with an average radius of 130 km was tracked from his genesis in the TT region on 14-October 2014 using the automated eddy tracking algorithm. This mesoscale eddy has travelled a distance of about 2580 km in
the ocean interior with an average translation velocity of 13.85 cm/s and passed south of the observation array moored in SR in the framework of collecting oceanographic data in the potential region for future DSM (see Figure10e).

The outer edges of the anticyclonic eddy with positive sea surface height anomalies higher than 10 cm reaches the SR on 05-April-2015. The ocean currents start tilting in northward direction to the North at 12-April and as the eddy approaches closer to the location of the mooring, the current direction reaches to its consistent northward direction in early May. The clear
impact of surface anticyclonic eddy is found in the deep layer of the ocean with a strong rotation of the dominant current direction from south to the north at all layers. The northward changes of current direction fits well with the clockwise rotation of anticyclonic eddy. The impact of maximum SSHA that occurs at day 19-April is observed when the current velocities reach their maximum strength and the current direction shows a strong deviation from its dominant soutward direction. The slightly stronger northward currents continuously last for 3 weeks until 18-May and the current direction returns to the main southward
direction afterward (see current direction after 20-May at Figure 10b-d). The impact of observed eddy is more attributed to the rotation of the dominant current direction than to current intensification. The current rose diagram of mean current speed for all Aquadopps is also generated to better indicate the rotation of dominant current direction while the mesoscale eddy passes the region (Figure10f). Nevertheless, the current velocities show values slightly larger than $4.5 \, \mathrm{cm \, s^{-1}}$ for a period of 4 weeks at the upper layers (compare the Figure10 b to d). The weak intensification observed in current velocity during May 2015 is
due to the large distance of the eddy center, which is about $1°$ away from the current meter array to the south which reduces the eddy impact on current velocity intensification. The lag-times for changes in current velocity and direction observed in our measurements are consistent with the interactions of hydrodynamic processes caused by the surface eddies.





**Figure 9.** Relationship between the Ocean Nino Index (ONI) based on the index 3.4 and a) EKE in the TT and PP gap wind regions, b) number of eddies, c) radius of eddies, d) vorticity of eddies, e) lifetime of eddies, f) intensity of eddies, all with lifetimes longer than 28 days.



Comparing the current magnitude at different depths shows a declining tendency suggesting that the feedback of the deep ocean to the surface eddies is attenuated with increasing depth. However, as indicated here, the eddy-induced hydrodynamic
influence on the observed near-bottom ocean current characteristics lasted for a month and it can be assumed that similar modulations of current properties can be seen for the entire water column.

The lagging feature of deep ocean current response to the passage of a surface eddy observed in this region is similar to the results showed by Zhang et al. (2014) in the South China Sea (SCS) where they found a lag of 12 days between current velocities at the deep layer and the surface geostrophic velocity. These authors also showed that there are longer lags between
the observed near bottom suspended sediment concentration and surface geostrophic velocities.

Despite some general similarities at responses of deep sea to surface mesoscale eddies at SCS and observed eddy at the SR, the lagging response and current magnitude intensifications are different. The different lagging response of bottom current properties to passing surface mesoscale can be attributed at first order to the different water depth and vertical stratification in the two basins. The measurements at SCS are taken at water depth of 2600 m which causes a faster transfer of EKE to the
deeper layers in this region. Moreover, the slower translation speed of ACE at SCS (10 cm/s) more likely causes higher energy uptake by deeper layer of ocean and results in a significant current intensification in SCS.

With respect to the hydrographic response of the deep ocean to the mesoscale surface eddies, several studies have shown that sediments in deep oceanic basins may be actively resuspended and redistributed where the bottom current regime is significantly enhanced due to eddy activity (Gardner and Sullivan, 1981; McCave, 1986; Isley et al., 1990). In the Gulf of
Lions, episodes of local sediment resuspension appear to occur for current speeds between 17 and 30 cm/s (Durrieu de Madron et al., 2017). Moreover, Gardner et al. (2017) showed that minimum current speeds required to resuspend material from the local seabed in western North Atlantic ocean is likely in the range of 10-20 cm/s which is observed to happen often and closely match with high deep EKE episodes. The eddy-induced current velocities observed between April and May 2015 in CCZ region reached only to 5-6 cm/s. This is below required bottom current velocities of 9-12 cm/s observed in laboratory experiments by
Gillard et al. (2019) with local sediment from the German license area in the CCZ. The optical sensors attached to the OBM did not record an increase of turbidity level while the surface mesoscale eddies passed the location of the moorings. Hence, the observed eddy-induced bottom current velocity at the bottom were not strong enough to resuspend naturally deposited deep sea sediment in investigated CCZ region. However, it can be assumed that freshly redeposited sediment from a mining-generated plume most likely requires lower shear stress at the seabed for resuspension which might be achieved even by the low EKE
bottom regime driven by a weak surface mesoscale eddy.

Our short-term observation suggests that eddy-driven impacts not only extends into the deep-sea benthic environment, but also depending on the eddy strength, eddy track and sea water hydrographic condition can modulate the hydrodynamics of the deep ocean environment which plays a vital role in the distribution of suspended sediment plumes from mining activities and their redeposition on the seafloor. Stronger bottom currents will inevitably drastically enlarge the spatial footprint of the mining-
induced sediment plume, however, sediment concentrations will decrease due to the increased dispersion. As a consequence, the resettled sediment blanket on the seafloor adjacent to the mined area will become thinner and more extended. Hence, eddies





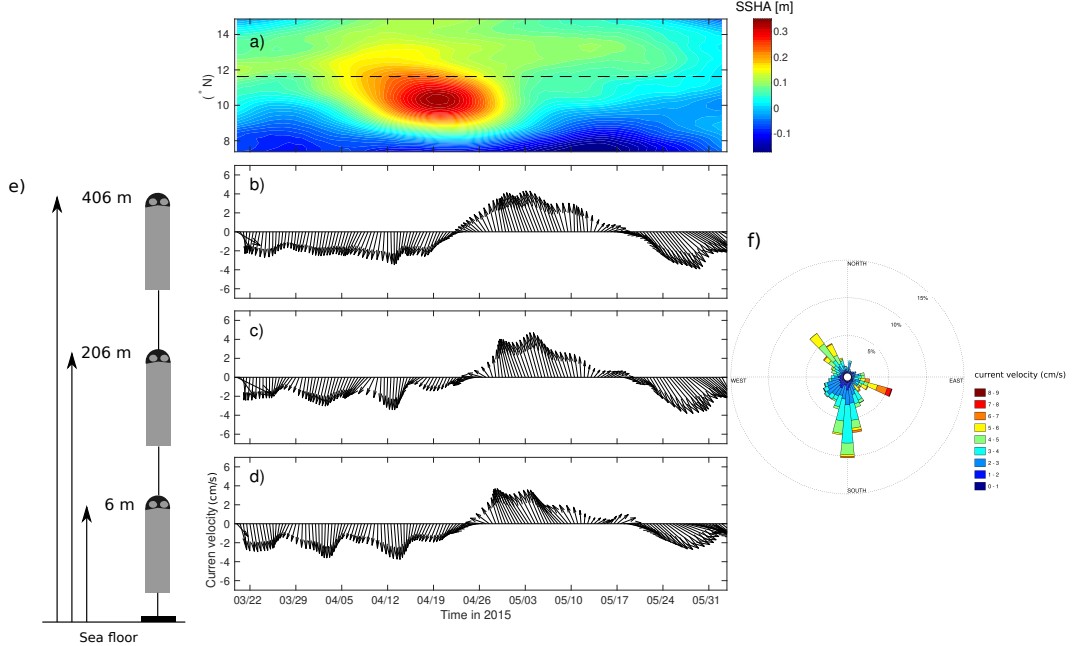

**Figure 10.** a) Temporal evolution of SSHA at a zonal transect at SR from 7° to 15°N from 20-March to 02-July 2015. Ocean current velocity recorded by an array of single-point Aquadopp current meters at 406 m above seafloor b), at 206 m above seafloor c), and at 6 m above seafloor d). These heights above seafloor are corresponding to 3788 m, 3971 m and 4180 m water depth. e) a schematic figure of the current meter array. The black dashed line in a) shows the latitude where the geographical center of the moorings is located. f) current rose diagram of mean current velocities of all Aquadopps during the same period. Current velocities are shown in cm/s and color corresponds to current speed classes. All current velocities are averaged over a half-day period and shown here.

will strongly alter the expected spatial footprint and severeness of the environmental impact. A key parameter for assessing the environmental impact of sediment plumes is the EKE.

## 4.2 Long term ocean current variability and its relationship to EKE at the region of gap winds

In this section we aim to assess the importance and linkage of sea surface mesoscale eddies which are generated at the vicinity of gap winds to the annual variability of deep ocean current properties in the SR. A combination of a relatively long-term deep ocean current measurements (three years between 2013 and 2016) at SR, a merged satellite SSHA data and an eddy-resolving global ocean reanalysis product are employed.



To investigate the interaction between the EKE in the region of the gap winds and in the SR approximately 2500 km away
from the coast, we computed the long term monthly mean EKE averaged over 24 years in both regions. The impact of high-
frequency wind events on the variation of EKE at the study site is limited to the period between late autumn and early spring
when higher EKEs extended into the ocean interior (Figure. 1b). A pronounced seasonal variability of EKE in the TT and PP
regions is observed, with slightly different timing despite the close geographical location of TT and PP (Figure. 11a). This is
due to the different mechanisms of wind generation and its development in these regions. The monthly mean EKE in the PP
gap wind region is generally higher, with a maximum in the winter ($> 800\,\mathrm{cm^2\,s^{-2}}$). A second maximum occurs in August
which is related to a local increase in wind velocity (Romero-Centeno et al., 2003). The level of EKE in the TT region is
marginally lower with an increase starting in November that lasts until the end of February (Figure 11a). Strong seasonality in
surface EKE at the ocean interior is also observed, although with a different timing (see the blue line in Figure. 11a ). Here, an
absolute maximum value of EKE occurs between April and June.

Unlike most of the eddies formed in the PP gap wind region which are observed to dissipate before they reach west of
110°W, most of the long-lived eddies with origin close to the TT region travel long distances into the ocean interior (see
Figure 7). Therefore, the variability of EKE in the ocean interior is suggested to be more susceptible to alteration by eddies
originating from the TT region. The correlation analysis of EKE shows a significant relationship with a correlation coefficient
of 81% between EKE at TT and SR with a time lag of 224 days. However, no significant correlation between EKE in the PP
region and the far offshore region was obtained. The time lag is in agreement with the time required for an ACE originating
in the TT region to propagate with an average translation speed of 12 cm/s to reach the SR. Therefore, we conclude that the
observed seasonal variability of EKE in the open ocean is primarily driven by the eddy-dependent anomalous transport of EKE
originating from the TT gap wind region.

To illustrate the relationship between passing surface eddies in the ocean interior and deep ocean current characteristics close
to the seafloor at a water depth of 4100 m, monthly mean current velocities obtained from the three OBM over a period of four
years from April 2013 to May 2016 were calculated (black line in Figure 11b). The qualitative feature of EKE seasonality at
the ocean surface is transferred through the entire water column to the bottom of the ocean with a time lag of about 1 month.
More intense current velocities with higher variability are observed between April and September at the seafloor (Figure 11b).
The average monthly current velocity reaches its maximum value 5.85 cm/s in May, indicating an increase in current velocity
from the mean value of 3.9 cm/s by almost 40%.

A more detailed analysis of the linkage between the ocean surface and the deep ocean is not possible without current
measurements throughout the water column. The evidence of a seasonal trend of deep ocean current velocities even over a
period of four years is, however, persuasive as it may invoke uncertainty on the essence of the observed signal and its intensity
for the longer periods. Nevertheless, we hypothesise that the observed seasonality is an important feature of deep ocean current
variability in this region. To examine this hypothesis, eddy-resolving global ocean reanalysis products with 1/12° horizontal
resolution and 50 vertical levels covering the time period between 1993 and 2016 were used (Drévillion et al., 2018). For
comparability, the model data were interpolated to the depth of the mooring measurements. Long-term reanalysis results
also indicate a clear seasonal behaviour in the variation of deep ocean current velocities, although with a generally weaker





magnitude (blue line in Figure 11b). Similar trends and the occurrence of a local peak in current velocity in May for both
observation and reanalysis of deep ocean current data confirms the observed evidences of deep ocean current seasonality and
the time lag in energy transfer from the ocean surface to the seafloor in this region.

    We also suggest that the strengthening of near surface ocean stratification and upward doming of isopycnals due to strong
rainfall and weak surface wind in the offshore ocean especially during April-May can prevent immediate transfer of kinetic
energy from the surface to the deeper layers and thus can be responsible for the observed lagging in the deep ocean current
velocities. Monthly mean upper ocean potential density stratification in the North Pacific, based on Argo data obtained between
2006 and 2016, indicates the highest stratification to occur in April in the regions close to the SR (Yamaguchi et al., 2019, see
Figure1).

    It is noteworthy to mention that seasonality in seafloor current strength may have a significant impact on the benthic com-
munity structure of the aphotic deep ocean where light intensity is assumed to be unimportant (Aguzzi and Company, 2010).
Biological monitoring in the Barkley Canyon, in the northeastern Pacific ocean, has shown that deep current seasonality can
be considered as a proxy for local seasonal drivers of species abundance due to its great influence on food availability and the
growth and reproduction cycle of benthic organisms (Doya et al., 2010).

## 5   Conclusion and future work

The mining operation in the deep sea will have environmental impacts due to the sediment plume and eventually redeposition
of suspended sediment in the water column. The insights gained from this study by analysis of time series collected from OBM
deployments that includes observation of current properties together with the SSHA data describes the eddy impact on deep
ocean current variability at far offshore regions. The results of this study aimed to assist deep ocean policy makers to adjust the
precautionary strategies to mitigate vulnerable influences of the future DSM in this region.

    Despite the general perception of a low energetic regime in deep ocean environment, our study has shown that the significant
seasonality of deep ocean currents and its diverse long-term variability are a natural feature of deep ocean environment of the
NETP which are closely related to the surface mesoscale eddy activity in this region. The passage of sea surface mesoscale
eddies may result in current velocity intensifications or a strong deviation from its dominant direction. In either case, deep
ocean current variability due to the passage of a surface mesoscale eddy can control the environmental impact of DSM in this
region (Aleynik et al., 2017) and might be able to mitigate the vulnerable impact in the near-field mining region by increased
suspended sediment dispersal as a consequence of stronger currents .

    Using SSHA data in NETP region from 1993 to 2016 and a geometry-based eddy detection algorithm developed by Nencioli
et al. (2010), significant differences between cyclonic and anticyclonic eddies in terms of eddy number, eddy radius, eddy
vorticity, eddy translation velocity and eddy lifetimes were detected. In total 6206 CEs and 5363 ACEs were identified during
24 years. For all detected eddies with lifetimes longer than one day the total number of cyclonic eddies developing off the coast
at Central America due to the gap wind activity exceeds that of anticyclonic eddies by a factor of 16%. However, for eddies
with lifetimes longer than 90 days there is a strong anticyclonic dominance in this region. The longest-lived cyclonic eddy





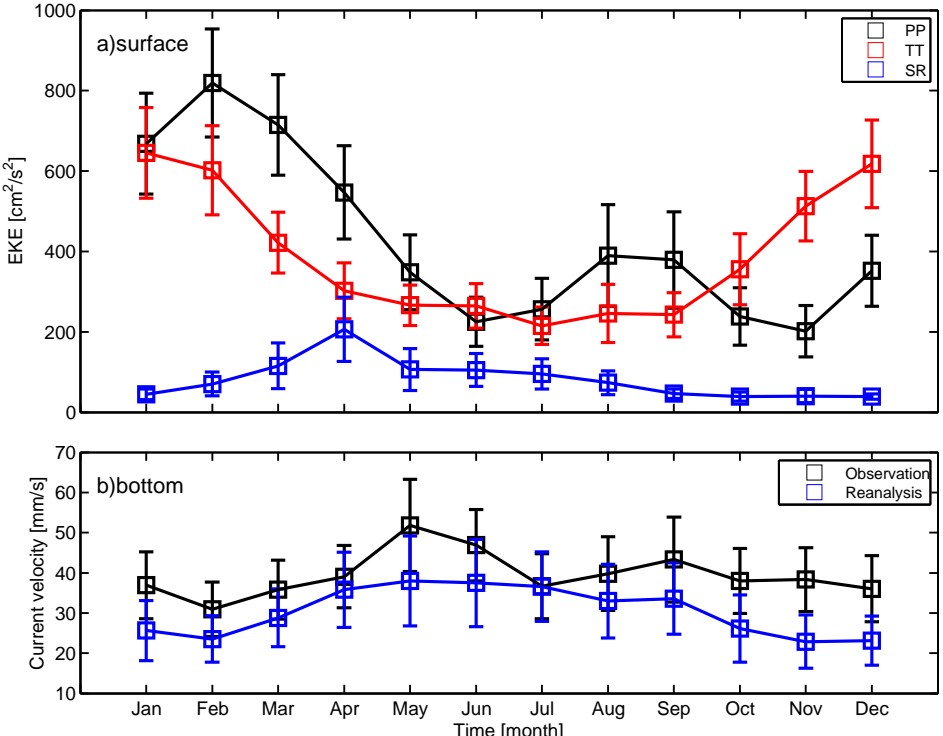

**Figure 11.** a) Annual cycle of monthly mean sea surface EKE averaged over 24 years for the Papagayo gap wind region (PP: averaged over 93.5-88.5°W, 10.5-12.5°N), the Tehuantepec gap wind region (TT: averaged over 100-95°W, 12.5-14.5°N) and the offshore study region (SR: averaged over 118-116°W, 10.5-12.5°N). b) Monthly averaged current magnitude (black line) at the seafloor of the SR from April 2013 until May 2016 obtained from three moored ADCPs. The blue line represents the same, but with data obtained from the reanalysis model and averaged from 1993 to 2016. Standard deviation with a 95% confidence level is shown as error bars. The error bars have been reduced to half of their size to avoid overlapping.

survived for a period of 113 days, whereas the longest-lived anticyclonic eddy survived for 321 days. No eddy with a lifetime longer than a year was observed in the NETP. The average size of ACEs reaches to 92 km, while CEs show a significantly smaller size with a radius of 84 km. The mean translation speed of long-lived eddies for ACEs and CEs are 12.5 and 6.8 cm/s

respectively.

The analysis of probability of eddy occurrence shows that long-lived eddies are principally generated near the coast, when locally intense gap winds in the TT and PP gulfs blow persistently into the ocean. Long-lived eddies with lifetimes in the category between 90 and 206 days are equally distributed between the TT and PP gap winds. In contrast, the TT gap wind is the only responsible agent for generation of eddies with lifetimes longer than 207 days. The eddy pathway is highly dependent

on the place of eddy formation. Our eddy track analysis shows that most of the eddies generated by PP travel in southwest





direction and terminate before they pass 110°W, while eddies formed in the vicinity of TT seems to travel longer distances in ocean interior in westward direction withouth moving meridionally after passing the EPR. The ACEs show faster translation velocity than the CEs regardless of their sourfce of generation. Moreover, it is found that ACEs are larger in size and intensity than the CEs.

All of the long-lived eddies were found to be nonlinear whith travel distances between 1500 km and 4570 km in the ocean interior which means that they may have a significant impact on anomalous transport of heat and salt while propagating into the ocean interior. Temporal elevation of EKE in the surface ocean interior at a potential future DSM site about 2500 km away from the coast are driven mainly by seasonal fluctuations of EKE in the TT gap wind region, with a time lag of about 7 months. The comparison of reanalysis data, current property measurements and satellite altimetry data shows an enhancement of deep

ocean currents with a lag of about three weeks in response to passing anticyclone mesoscale eddies at the ocean surface.

    On the interannual scale, characteristics of cyclonic eddies appear to be significantly related to different cycles of ENSO, whereas anticyclonic eddies are weakly or in most cases not related to ENSO.

    The process of vertical energy transfer throughout the ocean layer, including the impeditive effect of stratification, on lagging the response of deep sea currents to surface eddies is a complex issue that requires further analysis. Moreover, the mechanism

of merging eddies and the impact of merged eddies on anomalous transport of water masses in the ocean require more in detail studies in this region.

    Furthermore, deep-reaching eddies can have a significant influence on the settling velocity of marine snow and the alteration of the physical parameters of suspended sediment particles (sinking velocities, sediment transport, sediment resuspension) due to effects of current-induced aggregation and disaggregation processes. Such influences are tested in the laboratory (e.g.,

Gillard et al., 2019) but will also benefit from additional in situ observations as DSM trial become reality.

    The fundamental knowledge of eddy-related deep sea current variability and its linkage to the ocean surface processes are essential to inform the development of the precautionary approaches by the ISA to mitigate the effects of plume dispersion based on SSHA and surface mesoscale eddies in this region. It is, however, not clear yet that reduction of near field redeposition and increment of far field sediment dispersion which are the most possible impact of passing eddies on sediment distribution

are beneficial or detrimental for the deep sea environment. This study illustrates that other potential DSM regions around the world with a chance of passing surface mesoscale eddies due to their possible deep sea impacts require a larger attention to establish a dynamic regulatory framework for DSM operations based on the ocean surface eddy regime.

## 6   Acknowledgement

This study is supported financially by the German Ministry of Research through the MiningImpact project (grant nos. 03F0707A+B,

03F0708A) as a part of the Joint Programming Initiative of Healthy Seas and Oceans (JPI Oceans). The altimeter products were produced by Ssalto/Duacs and freely distributed by AVISO (www.aviso.oceanobs.com). The eddy resolving global ocean re-analysis products are available online at MERCATOR GLORYS12V1 (www.mercator-ocean.fr). The authors confirm that there are no known conflicts of interest associated with this publication.

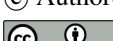



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
