# Peer review of "Evidence of eddy-related deep ocean current variability in the North-East Tropical Pacific Ocean induced by remote gap winds"

_Biogeosciences, 2020_

## Referee Comment (RC1) · Anonymous Referee #1 · 2 May 2020

Evidence of eddy-related deep ocean current variability in the North-East Tropical Pacific Ocean induced by remote gap winds

Kaveh Purkiani, André Paul, Annemiek Vink, Maren Walter, Michael Schulz, and Matthias Haeckel

General comments: In this study the impact of eddies on the variability of deep ocean currents is investigated in the northeast tropical Pacific Ocean by using a combination of satellite and in situ observations. The studied region is one of the most eddy-rich regions in the world ocean, where eddies are generated by remote wind gaps in the Sierra Madre mountains. This region is of special interest as the interest of deep-sea

mining is increasing. The main strength of this paper is to propose a thoroughly study of the main parameters of eddies and their spatiotemporal variability. My main concern is that the presentation of the analysis of the interaction between the EKE in the gap wind region and the EKE in the study region of the open ocean in not precise and that results are difficult to follow. A more carefully and detailed description of the analysis is lacking. I therefore recommend publication of this manuscript after major revision. I will leave my questions in the specific comments.

Specific comments:

L103: "July 2013 and April 2016": This period does not agree with the figure caption of figure 11b, which indicates to show data from April 2013 to May 2016. Please clarify.

L107: The mooring was deployed for three years between 2013 and 2016. Now you talk about four years ("60 min during the first year and 45 min during the following three years"). Please clarify.

L109: "20 March to 2 June 2015": This period does not agree with the figure caption of figure 10, which indicates a mooring period from 20 March to 2 July 2015. Please clarify.

L109: Please add the geographical location of the mooring.

L168: The swirl velocity is mentioned in chapter 3.2 as well as in chapter 3.5 "Translation speed and swirl velocity of eddies". Unfortunately, I could not find anything about the swirl velocity. I think it could be quite important concerning the lifetime of an eddy.

Figure 1: I would recommend to remove the EKE for the Atlantic Ocean in Figure 1 and 2. Figure 4 shows the zonal variability of meridionally averaged EKE and one might think that the EKE of the Atlantic was taken into account.

Figure 4: According to Figure 2 the regions TT and PP are at about 95°E and 85°E, respectively. Why is it shown at ∼98°E and ∼92°E in Figure 4?

L226: Why don't you choose the median eddy radius?

L235: There is no word about the swirl velocity, although it is even mentioned in the title. Please add information/estimates about the swirl velocity.

L267: It is not a surprising statement that the distance of eddies increases with their lifetime. Please rephrase or skip the sentence.

Figure caption of Figure 7: The last sentence is misleading. Please rephrase.

L275: I would recommend to calculate the annual mean for the period from July to June for each characteristic eddy parameter as El Nino/La Nina events show their peak during the turn of the year. Sometimes an El Nino year is followed by a La Nino year, which means that an annual mean from Jan-Dec would cancel the anomaly.

L287: I don't understand the last part of the sentence as the EKE at TT seems to be even more related to EL Nino events. Please clarify.

L323: I cannot see a tilting of the currents to a northward direction on 12 April, it seems to be rather on 22 of April in 406 m a.b. and on 25 April at 6m a.b..

L327: At the end of March as well as at the beginning of April, there is also a strong deviation of the current velocities. The reverse of the deep current from a southward to a northward direction occurs with three weeks after the time of maximum SSH showing a strong time lag between the passage of the eddy and the response of the deep current.

L393: Why do you calculate the correlation between the EKE at TT and SR from the annual cycle over 24 years and not from the time series for the whole period from 1993 to 2016?

L394: The lag of 224 is not clear to me. The peak in the vicinity of the TT region occurs in December. The maximum of the EKE in the SR occurs in April, which means a lag of 120 days. Please clarify.

L400: It is not clear to me, why monthly mean current velocities are used instead of the time series for the whole period of three years. Please explain.

L400: "four years from April 2013 to May 2016": It should be three years. Please correct.

L408: "four years". Should be three years.

Figure 11: Do you show northward velocity? In Figure 10 the current velocities show a southward current except for the passage of the ACE. Please clarify. Units are given in mm/s and in cm/s. Please, choose consistent units.

Technical corrections: L2: Better: "world ocean"

L10: typo: correct "that" into "than"

L32: typo: correct "kilimeters" into "kilometers"

L110: Usually, the figure number should be sorted by their order of appearance, which means that Figure 10 should be Figure 1.

L113: typo: correct "products" into "product"

Figure 1: The indication of a), b), c), d) is missing in the figures. It is not immediately clear which season is shown in which figure.

Caption of Figure 4: typo: delete one "period"

L241: The translation speed should be uniformly given to one decimal place.

Table 1: Please add the information in the caption, that this table only includes statistics about ACEs.

L254: The time interval of 37 days does not agree with the numbers in table 1.

Figure 7: The location of eddy generation is difficult to recognize, especially the ones that are coloured in yellow and orange.

Figure 8: typo at the label of the y-axis: correct "ifetime" into "Lifetime". Hyphens are missing at the label of the colorbar. Please add hyphens. Indication of a)-d) is missing in the figure caption. Please add a)-d).

L323: typo: correct "North" into "north".

L328: typo: correct "soutward" into "southward".

Figure 9: 9a shows different y-axis scales.

Figure caption of Figure 9: Please indicate that TT is black and PP is red in the figure caption.

Figure 10: Figure 10f is very small, hard to see and does not give any new information that cannot be obtained from Fig. 10a-d. I would recommend to drop Fig. 10f.

L431: typo: correct "describes" into "describe"

Figure 11: In Figure 9, the EKE in the TT (PP) gap wind regions is shown in black (red). In Figure 11 it is the opposite way round, which is confusing on first sight. Please plot the EKE of TT and PP in the same colour each.

―――――――――――――――――

---

## Referee Comment (RC2) · Anonymous Referee #2 · 2 May 2020

This is a study of mesoscale eddies and its effect on abyssal currents in the Northeastern Tropical Pacific (NETP). There are a number of papers on eddy variabilities and the effect of eddies on deep ocean currents, as noted by the authors. This study complements those studies by offering more detailed information about eddy properties and statistics, and identify the relationship between surface eddy activity and bottom current variability by using longer-period measurement. I would support publication given the following suggestions are considered. The revision would require efforts between minor and major revision.

1.     Section 3.1, the calculation of EKE is based on SSHA that is the de-

viation from the long-term mean. I think the "eddy" here is different from mesoscale eddies, as it includes seasonal and interannual variation, as well as mesoscale and submesoscale eddies. For example, there is a strong seasonal cycle in the mean circulations (e.g., Kessler 2006 The circulation of the eastern tropical Pacific: A review, Prog. Oceanogr., 69, 181–217. https://www.sciencedirect.com/science/article/abs/pii/S0079661106000310) and the EKE in calculated here includes those signals. Since the focus is on mesoscale eddies, a filter that also removes seasonal cycle and low-frequency variability seems more appropriate (see, e.g., Chelton et al. 2007; Liang et al. 2012). These two papers are in the reference list of the manuscript).

2. Section 3.7, there are two papers (Zamudio et al. 2001 ENSO and Eddies on the Southwest Coast of Mexico. GRL (https://agupubs.onlinelibrary.wiley.com/doi/pdf/10.1029/2000GL011814; and Zamudio et al. 2006 Interannual variability of Tehuantepec eddies. JGR-Oceans (https://agupubs.onlinelibrary.wiley.com/doi/full/10.1029/2005JC003182) about the relation between ENSO and eddy activities that were not cited. I would suggest the authors cite the papers and discuss how different the results in this study are from those papers.

3. When presenting statistics, I would suggest the authors add a significance level.

4. Section 4.1, the bottom current <10 cm/s seems to be weaker than previously reported value (>15 cm/s) by Adams et al. 2011 (Surface-Generated Mesoscale Eddies Transport Deep-Sea Products from Hydrothermal Vents). This may deserve some discussions.

---

## Author Comment (AC1) · 25 Jun 2020

We thank the reviewer for carefully reading our manuscript and helping us to clarify the presentation of our results a lot. All corrections, modifications and explanations are given in red color lines in the text of our manuscript as well.

Responses to Reviewer #1

L103: "July 2013 and April 2016": This period does not agree with the figure caption of figure 11b, which indicates to show data from April 2013 to May 2016. Please clarify.

Done. Our data presented in this study covers the period from April 2013 to May 2016.

[Figure]

Correction is added to the text.

L107: The mooring was deployed for three years between 2013 and 2016. Now you talk about four years ("60 min during the first year and 45 min during the following three years"). Please clarify.

The mooring was indeed deployed for three year the typo is corrected. "60 min during the first year and 45 min during the following two years".

L109: Please add the geographical location of the mooring.

Done, more additional information is added to the text: At a water depth of ca. 4100 m, three moorings were deployed 8 km apart at the vertices of an equilateral triangle with geographical coordinates of (11°51.11′N, 116°58.43′W), (11°48.30′N, 116°59.36′W) and (11°53.19′N, 117°00.48′W).

L168: The swirl velocity is mentioned in chapter 3.2 as well as in chapter 3.5 "Translation speed and swirl velocity of eddies". Unfortunately, I could not find anything about the swirl velocity. I think it could be quite important concerning the lifetime of an eddy.

The average surface swirling velocities (V$\theta$) of the eddies increase outward and reaches values of around 20 cm/s and 10 cm/s at the edges of the eddies for ACEs and CEs respectively. The nonlinearity parameter of eddies, which is characterized by the ratio of swirl velocity to translation velocity (V$\theta$/VT), is calculated. Most of the eddies of from both types in this region indicate a significant degree of nonlinearity (V$\theta$/VT>1), implying that eddies can maintain a coherent structure, which may isolate the interior water mass without interaction with ambient water while propagating in the ocean. Similar to previous study (Stramma et., al 2014 and Czeschel et al., 2018) in the South Pacific Ocean, the activity of nonlinear long-lived eddies in this region may result in the large-scale anomalous water mass distribution in this region.

Figure 1: I would recommend to remove the EKE for the Atlantic Ocean in Figure 1 and . Figure 4 shows the zonal variability of meridionally averaged EKE and one might

think that the EKE of the Atlantic was taken into account.

Done! In the new figures the regions in the Atlantic Ocean are masked out.

Figure 4: According to Figure 2 the regions TT and PP are at about 95 âŮę E and 85 âŮę E, respectively. Why is it shown at âĹij98 âŮę E and âĹij92 âŮę E in Figure 4?

Figure 4 does not show the geographical location of TT and PP gap winds, but the zonal variation of laterally averaged EKE in the ocean with the local maximums driven by TT and PP gap winds. The caption of Figure 4 is corrected for more clarification as below. "The dashed lines show the EKE of the ocean circulation in the open ocean and two local maxima driven by TT and PP gap winds respectively".

L226: Why don't you choose the median eddy radius?

The eddy radius exhibits a skewed distribution due to long-lived mesoscale eddies. Using a median instead of a mean should therefore result in a more robust estimate of the center of the distribution than using the mean. We believe that using the median instead of mean would pull out the outliers in the tails of the histogram distribution. This outlier data are mainly the long-lived mesoscale eddies with the main focus of this study. Therefore, the mean is chosen for statistical analysis.

L235: There is no word about the swirl velocity, although it is even mentioned in the title. Please add information/estimates about the swirl velocity.

Additional information is added in section 3.5.

L267: It is not a surprising statement that the distance of eddies increases with their lifetime. Please rephrase or skip the sentence.

Done! It is removed.

Figure caption of Figure 7: The last sentence is misleading. Please rephrase.

Done! The lifetimes of detected eddies are divided to six classes with the length of

38 days for each class. Tracks of eddies are shown in colors corresponding to each lifetime class.

L275: I would recommend to calculate the annual mean for the period from July to June for each characteristic eddy parameter as El Nino/La Nina events show their peak during the turn of the year. Sometimes an El Nino year is followed by a La Nino year, which means that an annual mean from Jan-Dec would cancel the anomaly.

This idea has been earlier tested. Changing the averaging period did not help to improve the statistical correlation between ONI and eddy characteristic parameters. By taking the period from July to June for eddy characteristics, some times even less significant correlation was obtained.

L287: I don't understand the last part of the sentence as the EKE at TT seems to be even more related to EL Nino events. Please clarify.

Some more discussion is added to this section. Please see lines 300-30

L323: I cannot see a tilting of the currents to a northward direction on 12 April, it seems to be rather on 22 of April in 406 m a.b. and on 25 April at 6m a.b..

It is corrected. We have earlier defined the tilting as a time in which zonal current velocity increases enough high to decline the southward current direction. However, we it more informative if the northward tilt was mentioned. The ocean currents tilt into northward direction at 22-April and 25-April at 406 m and 6 m above the seafloor respectively.

L327: At the end of March as well as at the beginning of April, there is also a strong deviation of the current velocities. The reverse of the deep current from a southward to a northward direction occurs with three weeks after the time of maximum SSH showing a strong time lag between the passage of the eddy and the response of the deep current.

The deep sea in this region is indeed a dynamic environment, characterized by daily

to seasonal variations in current properties. We agree that variations in the current velocity occur at the end of March and early April. However, during this period no significant variation in the current direction was observed. An analysis of sea-surface height anomalies indicates that the weak current variation signal cannot be related to the anomalies driven by the passage of a mesoscale eddy as no significant SSHA is observed in this period. We therefore summarize that as the current variation signal is only limited to the lower current meters at 6 mab and 206 mab, this must be due to interaction of deep sea current and bottom topography in this region, which could not develop at higher levels above the seafloor (e.g., 406 mab).

L393: Why do you calculate the correlation between the EKE at TT and SR from the annual cycle over 24 years and not from the time series for the whole period from 1993 to 2016?

This is modified based on the daily time series for the whole period. More explanation is given below.

L394: The lag of 224 is not clear to me. The peak in the vicinity of the TT region occurs in December. The maximum of the EKE in the SR occurs in April, which means a lag of 120 days. Please clarify.

Analyzing the lag correlation between EKE at the location of the gap winds with EKE at SR based on daily time series led to a reduced and correction of the time lag to 165 days with a maximum correlation of 83% between EKE at TT and SR. No significant correlation was again found between PP and SR. The time lag between EKE at the gap winds and EKE at SR is now consistent with the required time for a long-lived ACE with the average translation velocity of 16.9 cm/s (see table 1) to travel a distance of 2400 km from TT gap winds to the SR region.

L400: It is not clear to me, why monthly mean current velocities are used instead of the time series for the whole period of three years. Please explain.

First of all we shall mention again that the black line in the Fig 11b is a combination of three moorings. As we have focused on the large scale responses of SR hydrodynamic properties as well as mesoscale eddies to the gap winds, analyzing the small-scale temporal variation of deep-sea current properties is not in the scope of this study.

Besides, the use of monthly mean current velocities is more consistent with the presentation and derivation of EKE from long-term surface data at different locations as well as the long-term reanalysis products for the deep sea.

By providing our data as monthly averages, we hope to contribute to reducing the environmental impact of deep-sea mining by broadening the knowledge of temporal variability of ocean current properties in the deep sea environment.

L400: "four years from April 2013 to May 2016": It should be three years. Please correct.

Done!

L408: "four years". Should be three years.

Done!

Figure 11: Do you show northward velocity? In Figure 10 the current velocities show a southward current except for the passage of the ACE. Please clarify. Units are given in mm/s and in cm/s. Please, choose consistent units.

Figure 11b shows the monthly average of the ocean current speed (Vel = (U2+V2)0.5) . In this figure more emphasis is given to the fact that long-term current observation as well as reanalysis products are able to indicate seasonal anomalies even in the deep sea when mesoscale eddies cross this region. As it was mentioned earlier in the caption of this figure no change is made.

Units are changed to cm/s in the figure.

Technical corrections: L2: Better: "world ocean"

Done!

L10: typo: correct "that" into "than"

Done!

L32: typo: correct "kilimeters" into "kilometers"

Done!

L110: Usually, the figure number should be sorted by their order of appearance, which means that Figure 10 should be Figure 1.

Done!

L113: typo: correct "products" into "product"

Done!

Figure 1: The indication of a), b), c), d) is missing in the figures. It is not immediately clear which season is shown in which figure.

The original figure (please see attached Figure 1) had all the indications. This must been a technical issue. In the new figure the indications from a) to d) are better positioned and shown.

Caption of Figure 4: typo: delete one "period"

Done!.

L241: The translation speed should be uniformly given to one decimal place.

Done!. In the entire text the translation velocities are given to one decimal.

Table 1: Please add the information in the caption, that this table only includes statistics about ACEs.

Done!.

Figure 7: The location of eddy generation is difficult to recognize, especially the ones that are coloured in yellow and orange.

We have tried with other colors and markers, but they were covered anyways and it is difficult to recognize them as there are more eddies specially in the shorter lifetime classes. Therefore, we kept the figure as it was.

Figure 8: typo at the label of the y-axis: correct "ifetime" into "Lifetime". Hyphens are missing at the label of the colorbar. Please add hyphens. Indication of a)-d) is missing in the figure caption. Please add a)-d).

Done! All suggestions have been followed. Similar to Fig 1, the indication of a) to d) was missing in the figure after compiling in the journal template. The original figure (please see attached figure 8) had all indications. The issue will be discussed with the technical support of the journal.

L323: typo: correct "North" into "north".

Done!

L328: typo: correct "soutward" into "southward".

Done!

Figure 9: 9a shows different y-axis scales.

Done! Scales in right axis are edited.

Figure caption of Figure 9: Please indicate that TT is black and PP is red in the figure caption.

The black and red dots in a) refer to EKE at TT and PP, and in b) to f) refer to eddy characteristics of ACEs and CEs respectively.

Figure 10: Figure 10f is very small, hard to see and does not give any new information that cannot be obtained from Fig. 10a-d. I would recommend to drop Fig. 10f.

Done! Fig. 10f is removed. The caption is edited accordingly.

L431: typo: correct "describes" into "describe"

Done!

Figure 11: In Figure 9, the EKE in the TT (PP) gap wind regions is shown in black (red). In Figure 11 it is the opposite way round, which is confusing on first sight. Please plot the EKE of TT and PP in the same colour each.

Done! It is all corrected. The color of TT and PP follows the same as what is presented in Figure 9.

———————————————————

[Figure]

[Figure]

[Figure]

**Fig. 2.**

[Figure]

[Figure]

Fig. 3.

[Figure]

**Fig. 4.**

[Figure]

**Fig. 5.**

[Figure]

[Figure]

**Fig. 6.**

[Figure]

**Fig. 7.**

[Figure]

**Fig. 8.**

[Figure]

**Fig. 9.**

[Figure]

**Fig. 10.**

[Figure]

Fig. 11.

[Figure]

---

## Author Comment (AC2) · 25 Jun 2020

We thank the reviewer for carefully reading our manuscript and helping us to clarify the presentation of our results a lot. All corrections, modifications and explanations are given in red color lines in the text of our manuscript as well.

Responses to Reviewer #2 1. Section 3.1, the calculation of EKE is based on SSHA that is the deviation from the long-term mean. I think the "eddy" here is different from mesoscale eddies, as it includes seasonal and interannual variation, as well as mesoscale and submesoscale eddies. For example, there is a strong seasonal cycle in the mean circulations (e.g., Kessler 2006 The circulation of the eastern tropical Pacific: A review, Prog. Oceanogr., 69, 181–217. https://www.sciencedirect.com/science/article/abs/pii/S0079661106000310) and the EKE in calculated here includes those signals. Since the focus is on mesoscale eddies, a filter that also removes seasonal cycle and low-frequency variability seems more appropriate (see, e.g., Chelton et al. 2007; Liang et al. 2012). These two papers are in the reference list of the manuscript).

As we are interested in the interannual variability of mesoscale eddies, related to ENSO events, we did not apply any filters to the SSHA (such as done in Liang et al.2012) to keep seasonal cycle and low-frequency variability. The SSHA are the daily mean data with removed any signals higher than a day originally.

2. Section 3.7, there are two papers ( ENSO and Eddies on the Southwest Coast of Mexico.GRL (https://agupubs.onlinelibrary.wiley.com/doi/pdf/10.1029/2000GL011814; and Zamudio et al. 2006 Interannual variability of Tehuantepec eddies. JGR-Oceans (https://agupubs.onlinelibrary.wiley.com/doi/full/10.1029/2005JC003182) about the relation between ENSO and eddy activities that were not cited. I would suggest the authors cite the papers and discuss how different the results in this study are from those papers.

The impact of ENSO events on the intensity and frequency of northerly winds at TT is more complicated and has been addressed in previous studies (e.g., Romero-Centeno et al., 2003; Zamudio et al., 2006). In contrast to the La Niña years during which winds are significantly weaker and the occurrence of northerly winds is significantly rarer, during El Niño years the more frequent occurrence of strong northerly winds is restricted only to May and September.

Despite the lack of a significant increase of suitable winds for mesoscale eddy formation at TT during El Niño years, a larger number of mesoscale eddies in agreement to our results is reported in this region in previous studies (Zamudio_2001, Palacios2005, Zamudio_2006). Thus, in contrast to the initial hypothesis, eddy formation and its interannual variability in the TT region cannot be solely explained by strong and intermittent wind events. The analysis of a high-resolution ocean model forced by ECMWF meteorological data from this region shows that an increase in propagating downwelling coastally trapped waves (CTW) during El Niño years plays a crucial role in the modulation and generation of TT eddies (Zamudio et al., 2006). While the CTWs propagate along the coast of Central America and Mexico, a strong horizontal and vertical shear of the horizontal velocity is generated, which can trigger barotropic and baroclinic instabilities. The breaking of long-wavelength CTW meanders generates mesoscale eddies in this region (Zamudio et al., 2006).

Please see section 3.7 in the MS for the further discussions.

3. When presenting statistics, I would suggest the authors add a significance level. For testing the hypothesis of no correlation the P-values are calculated. All the P-values for the Pearson correlation coefficients of TT and SR EKE are too small, indicating a significant correlation. This is added to the text at line 439. For the section 3.7 as we have used the annual mean of ONI and eddy characteristics, the calculation of P-values is most likely unrealistic.

4. Section 4.1, the bottom current <10 cm/s seems to be weaker than previously reported value (>15 cm/s) by Adams et al. 2011 (Surface-Generated Mesoscale Eddies Transport Deep-Sea Products from Hydrothermal Vents). This may deserve some discussions.

Thank you very much for pointing us to this interesting paper. In general, the paper supports most of our findings as well as the possibility of a seasonal cycle in deep-sea currents. The following discussion is added to our manuscript at line 370-390 in red.

The impact of surface eddies on deep sea current velocities in the NETP has been addressed earlier by Adams et al. (2006). The current velocities measured at depth of 2430 m from May to June 2007 at 9°50.0N, 104°17.4W show deep sea current velocity exceeding 15 cm/s during the sea surface height anomaly which is almost three times

faster compared with the mean current speed of 5.5 cm/s at this region. The stronger current velocities observed in this study as compared to our measurements are likely due to the more significant impact of eddy on the deep sea due to shorter distance of eddy center from the mooring arrays, while our moorings are almost located at a distance of about 100 km away from the eddy center. Besides, the larger EKE of the young eddy and the shallower water depth in this region could be another resons for larger deep-sea current velocities. Due to the geographical location of the observations in their study, eddies in their stable life stage with relatively larger EKE content reach this region first. The shallower water depth at this region may also cause less energy dissipation in the ocean layers thus currents at deep sea content larger EKE. The lagging feature of deep ocean current response to the passage of a surface eddy observed in this region is similar to the results showed by Zhang et al. (2014) in the South China Sea (SCS) where they found a lag of 12 days between current velocities at the deep layer and the surface geostrophic velocity. The longer time lag observed by Zhang et al. (2014) can be related to the different water column stratification in SCS. These authors also showed that there are longer lags between the observed near bottom suspended sediment concentration and surface geostrophic velocities. Adams et al. (2006) also found that near-bottom current intensification is lagged the surface height anomalies by 8 days. Considering the water depth of approximately 2400 m in this study and 4100 m at the SR, the longer time lag of 15 days in our measurements between surface anomalies and the observed responses of the deep sea is shown to be linearly related to the water depth.

---

## Referee Report (RR1)

**Evidence of eddy-related deep ocean current variability in the North-East Tropical Pacific Ocean induced by remote gap winds**

**Kaveh Purkiani, André Paul, Annemiek Vink, Maren Walter, Michael Schulz, and Matthias Haeckel**

The authors have addressed all of my concerns adequately. Apart from some minor corrections I recommend accepting this paper in its present form.

Caption of figure 7: typo: Correct "divided in to six classes" into "divided into six classes".

Caption of Figure 8: The indications a)-d) are still missing in the figure **caption.** (No need for discussion with the technical support…)

L351-353: The authors refer to Figure 10f in the text, which is now removed from the manuscript.

---

## Author Response (AR2)

We thank the reviewer for carefully reading our manuscript and helping us to clarify the presentation of our results a lot. All corrections, modifications and explanations are given in red color lines in the text of our manuscript as well.

Responses to Reviewer #1

Caption of figure 7: typo: Correct "divided in to six classes" into "divided into six classes".

It is corrected.

Caption of Figure 8: The indications a)-d) are still missing in the figure caption. (No need for discussion with the technical support…)

The caption is corrected.

L351-353: The authors refer to Figure 10f in the text, which is now removed from the manuscript.

The text is removed from the MS.

Responses to Reviewer #2

I am mostly satisfied with the revision, except that I don't fully agree with the authors' response to my first suggestion in my first review. I would be happy to recommend publication if the authors could make the suggested clarification.

The following clarification is added in section 3.1.

[revised manuscript text omitted]